# A Reliable Method for Determining the Degree of Orientation of Fibrous Foods Using Laser Transmission and Computer Vision

**DOI:** 10.3390/foods12193541

**Published:** 2023-09-22

**Authors:** Jingpeng Li, Xu Xia, Cuizhu Shi, Xiaoqing Chen, Hao Tang, Li Deng

**Affiliations:** Key Laboratory of Agricultural and Animal Products Storage and Processing of Guizhou Province, School of Liquor and Food Engineering, Guizhou University, Guiyang 550025, China; jpli3@gzu.edu.cn (J.L.); x202210141014@163.com (X.X.); 15086247051@163.com (C.S.); 15120338405@163.com (X.C.); tanghao114665@163.com (H.T.)

**Keywords:** degree of orientation, image processing, laser transmission, computer vision, fibrous foods, method development, method verification, extruded jerky

## Abstract

The degree of organised alignment of fibre structures, referred to as the degree of orientation, significantly influences the textural properties and consumer acceptance of fibrous foods. To develop a new method to quantitatively characterise the fibre structure of such foods, a laser transmission imaging system is constructed to capture the laser beam spot on a sample, and the resulting image undergoes a series of image processing steps that use computer vision to translate the light and dark variations of the original images into distinct ellipses. The results show that the degree of orientation can be reasonably calculated from the ellipse obtained by fitting the outermost isopixel points. To validate the reliability of the newly developed method, we determine the degree of orientation of typical fibrous foods (extruded beef jerky, pork jerky, chicken jerky, and duck jerky). The ranking of the measured orientation agrees with the results of pseudocolour maps and micrographs, confirming the ability of the method to distinguish different fibrous foods. Furthermore, the relatively small coefficients of variation and the strong positive correlation between the degree of organisation and the degree of orientation confirm the reliability of this newly developed method.

## 1. Introduction

A well-organised fibre structure is commonly found in the internal tissues of fibrous foods such as jerky, plant protein meat, sugar cane, mushrooms, and hand-torn bread [1,2,3,4]. Such fibre structure has a significant influence on the textural properties of fibrous foods [5,6], which in turn influences consumer acceptance and preference [7]. Therefore, the quantitative description of the anisotropic properties in these fibrous foods is of considerable importance for food quality control.

Currently, the methods available to describe the characteristics of fibrous foods are limited. Zhang et al. [8] evaluated the fibrousness of extruded peanut proteins by analysing the ratio of shear forces in the perpendicular and parallel directions of the fibre arrangement using texture profile analysis. Samard and Ryu [9] also used texture profile analysis to investigate the cutting resistance of both transverse and longitudinal sections of meat analogues and meat samples. Furthermore, Dekkers et al. [10] analysed the fibrous texture formation process in concentrated soy protein isolate–pectin blends based on their viscoelastic properties. In addition, Krintiras et al. [11] reported the number of fibre layers and the radians of the orientation distribution for fibres using spin-echo small-angle neutron scattering. The above methods mainly rely on the textural–mechanical properties and micrographs of samples to study their fibre structures, making it difficult to describe the nature of the fibre structure quantitatively. In addition, the samples are usually destroyed, resulting in an inevitable waste of raw materials.

At the level of molecular structure, the essence of fibre structure formation is attributed to the ordered arrangement of macromolecules along a particular direction. According to the definitions in the field of polymer physics [12], orientation is the phenomenon whereby macromolecular chains, chain segments, or microcrystals are ordered in a certain direction under the influence of an external field (shear force, tensile force, etc.). The degree to which the oriented macromolecules, chain segments, etc. are neatly aligned along the fibre axis is defined as the degree of orientation. Commonly used methods to determine the degree of orientation of fibrous foods include wide-angle X-ray diffraction [13], birefringence measurements [14], ultrasonic wave propagation [15], polarised Fourier transform infrared spectroscopy [16], etc. However, these methods generally use polymeric materials produced by monomeric polymerisation only, whereas the main components of food samples are much more complex and heterogeneous, usually containing non-uniformly distributed proteins, sugars, fats, vitamins, dietary fibres, etc. The complexity of the interface among these different components makes it extremely difficult to apply these methods in the field of food science. It is therefore necessary to develop a reliable method for determining the degree of orientation of the fibre structure. 

Laser technology is a valuable tool in various research areas due to its advantageous properties, such as high brightness, good directionality, excellent monochromaticity, and high coherence [17]. These properties make lasers an excellent signal source for non-destructive determination, helping to reveal the inherent physical and chemical properties of fibrous food samples [18,19,20]. According to the principles of laser transmission, when laser light is transmitted at a specific angle to fibre samples, it undergoes a series of random reflections, refractions, scattering, and absorption within the fibre structure. The outgoing light after these complex interactions can be collected at different distances from the point of incidence of a laser source. This light contains a wealth of internal information about the fibre structure that is worthy of further analysis [21,22]. In particular, when a laser is irradiated from the surface of a fibre sample, the light tends to be transmitted along the direction of the fibres, resulting in the image spot of the outgoing light appearing as an ellipse.

Computer vision technology focuses on how computers can gain a high level of understanding from digital images or video, including algorithms for capturing, processing, and analysing digital images. For example, computer vision has been widely used to extract colour, texture, shape, and size features in food products for classification and identification purposes [23,24,25,26,27]. To extract the information that reflects the internal structure of the sample, the laser image can be captured using a digital camera instead of human vision, and then the laser image can be denoised, enhanced, and identified step by step using a computer. The combined use of laser transmission and computer vision can quickly extract digitised information from the laser spot image that is normally ignored or invisible to the human eye. In this case, the reliability of obtaining degree-of-orientation data is significantly improved to meet the practical requirements of the food industry. Ranasinghesagara et al. [28] developed a non-destructive photon migration method to visualise the degree of fibre formation and fibre orientation in meat analogues using a laser scanning system. Inspired by this idea, we aimed to develop a reliable method to determine the degree of orientation of fibrous foods.

As a new method for determining the degree of orientation of fibrous foods, its reliability needed to be validated. In accordance with the commonly used methods for describing the organisational characteristics of fibrous foods [29,30], scanning electron microscopy (SEM) and a texture analyser were used to verify the rationality of the degree of orientation trends of different fibrous samples. At the same time, pseudocolour maps extracted from the original laser images were used to validate the rationality of this method. 

As shown by the results of our previous study [31], high-protein-extruded products have pronounced fibrous structural characteristics, which have a significant impact on consumer acceptance and preference, especially for dried meat products. This result highlights the need to determine the inherent degree of orientation in this specific food category. Therefore, extruded pork jerky with a relatively well-organised fibre structure is used as a typical sample in the method development experiments, and additional meat products, including extruded beef jerky, chicken jerky, and duck jerky, are used in the subsequent verification experiments.

The objectives of this study are twofold: (1) to develop a reliable method for determining the degree of orientation of fibrous foods using laser transmission and computer vision, and (2) to verify the reliability of the newly developed method by comparing the orientation results of different extruded jerky products with their pseudocolour maps of laser spots, microstructural, and textural properties, with particular emphasis on the commonly used index for fibrous foods, called degree of organisation. The development of this method has a wide range of potential applications for other fibrous foods and has several advantages, including reliability, speed, non-destructiveness, etc. These attributes represent promising potential for future implementation in the field of on-line detection development.

## 2. Materials and Methods

### 2.1. Materials

Pork hindquarter, beef hindquarter, chicken thigh, and duck leg were all purchased from Walmart supermarket (Guiyang, China). All meat materials were stored in the freezer at 4 °C.

### 2.2. Sample Preparation

Dried meat products, namely pork jerky, beef jerky, chicken jerky, and duck jerky, were produced using extrusion technology. Specifically, the raw materials were subjected to a stepwise pre-treatment of pre-cooking, cutting, frying, drying, and stretching. Soya bean isolate protein and potato starch were then added at 10% and 8% of the wet weight of the raw materials. The final moisture content of the raw material was maintained at around 42%. The barrel temperatures of the twin screw extruder (DS32-II, Saixin Machinery Co., Ltd., Shandong, China) were kept constant at 75 °C (zone I), 133 °C (zone II), and 148 °C (zone III). The pre-treated raw materials were fed into the compatible screw feeder, and the experiments were conducted at a screw speed of 144 rpm and a feed rate of 158 rpm. All jerky samples were dried to a moisture content of 30%, after which the samples were cut to a size of 2 cm (length) × 2 cm (width) × 0.3 cm (thickness) for use. At least 10 samples of each material were prepared for the following data determination experiments.

### 2.3. Construction of the Laser Transmission Imaging System

The laser transmission imaging system was mainly composed of a *He-Ne* laser transmitter (HW-HN, Shenzhen Infrared Laser Technology Co., Ltd., Guangdong, China), a lifting platform (Botou Xingjiu Experimental Instrument Factory, Hebei, China), a neutral filter (HW-450, Shenzhen Infrared Laser Technology Co., Ltd., Guangdong, China), a digital camera (PowerShot A610, Hitachi High-Technologies International Trading Co., Ltd., Oita, Japan), a sample holder (Leiruo Instrumentation Technology Co., Shanghai, China), and a computer (XiaoXin Air14, Lenovo Co. Ltd., Beijing, China). The laser transmission imaging system was carefully adjusted to ensure alignment. The main components of the laser transmission imaging system are shown in Figure 1.

### 2.4. Laser Spot Image Acquisition Process

Figure 2 shows the laser spot image acquisition process using the constructed laser transmission imaging system. The fibre samples were irradiated by a *He-Ne* laser transmitter with a power of 2 mW and a wavelength of 632.8 nm. The laser beam passed through a neutral filter before reaching the fibre specimen, and the angle of incidence was set at 45°. This angle was further checked using a digital angle ruler (DL305200, Deli Group Co., Ltd., Zhejiang, China). To ensure that the red laser spot was in the position where the degree of orientation was to be determined, the specimen was placed parallel to the x-axis direction, and then the specimen was carefully moved along the y-axis. The laser spot images were captured using a digital camera with an aperture of F/5.6 and an exposure time of 1/800 s. The captured laser spot images were then transferred to the computer, and the images were processed and analysed by writing programs using MATLAB R2019a (The MathWorks Inc., Natick, MA, USA).

### 2.5. Processing Steps for Laser Spot Images

When the laser irradiates a rough sample plane, the high degree of laser coherence leads to the formation of laser scatter [32]. In the process of fitting the edges of a laser spot image, the spots have unpredictable positions on the image, and the spots in the unfitted region are considered to be noise [33,34]. The presence of noise degrades the image quality and affects the subsequent ellipse fitting process, so the noise on the laser spot image must be removed in advance.

The processing steps for laser spot images and the changes to the images during the process are shown in Figure 3. The specific operating methods and corresponding effects of each image processing step are illustrated in Section 2.5.1, Section 2.5.2, Section 2.5.3, Section 2.5.4 and Section 2.5.5.

#### 2.5.1. Gaussian Filters Denoising

Gaussian filters are low-pass filters based on a two-dimensional Gaussian function distribution principle [34,35]. The Gaussian filter is able to reduce noise while preserving the important details of the original image. This is achieved by assigning lower weights to locations further from the centre of the filter, as illustrated by the 5 × 5 Gaussian filter template shown in Figure 4a. The pixel values of the image before and after the 5 × 5 template processing are shown in Figure 4b,c, respectively. Thus, the noise in the original laser spot images was removed using 5 × 5, 7 × 7, and 9 × 9 Gaussian filter templates, and the optimal template was selected by comparing the noise reduction effect.

#### 2.5.2. Edge Enhancement

The image edges contain the most basic image shape feature parameters, and an enhanced denoising process of the laser spot image edges helps to identify the image edge features [36]. First, the edge filter operator Sobel in the MATLAB toolbox was used to detect the edges of the laser spot image, and then the outer edge pixel values were copied, and the corresponding data were extended to the periphery of the image, resulting in an edge enhancement effect [37]. A 3 × 3 edge filter template is used as an example to illustrate this process (Figure 5). Figure 5a shows the pixel values of the image before edge enhancement. Figure 5b,c depicts schematic illustrations of the results of edge enhancement at the top and right edges of the image, respectively. The desired edge enhancement effect for the whole image can be achieved by repeating the above operation around the edges of the image. In this way, the effects of 3 × 3 and 5 × 5 edge enhancement filter templates on the edges of the laser spot image were compared, and the optimal filter template was selected for further study.

#### 2.5.3. Morphological Smoothing

The edge-enhanced images may not have very smooth connections between pixels at the edges, which need to be further adjusted by morphological smoothing operations [38,39]. The morphological operations include expansion, erosion, open, and close operations. Among them, expansion and erosion are used as basic operations. Usually, expansion and erosion are combined. When erosion is followed by expansion, it is defined as an open operation, and when expansion is followed by erosion, it is defined as a close operation [40]. The morphological closing operation is a desirable algorithm when dealing with laser spot images, as this operation does not change the area of the image. Figure 6a shows the original image where the notches represent the discontinuous parts of the pixels. The discontinuities in the image are basically continuous after the expansion operation (Figure 6b), and the image boundary becomes much smoother after the initial expansion and subsequent erosion process (i.e., morphological closing operation), as shown in Figure 6c.

#### 2.5.4. Binarisation Treatment

After the morphological smoothing operations described above, the images need to be binarised. The pixel values contained in the binarised image are 0 and 1, where 0 and 1 represent the black and white background parts of the image, respectively. Since the junction of the white and black backgrounds is considered to be the image edge, it is easy to find the image edge coordinates for the final purpose of fitting an ellipse.

#### 2.5.5. Ellipse Fitting

The laser spot image after the above processing is ready for ellipse boundary fitting. The bwboundaries algorithm in the MATLAB software R2019a was used to track the binarised image spot boundaries to locate the edge coordinates, and then the least squares method was used to complete the outermost ellipse fitting process.

### 2.6. Method for Calculating the Degree of Orientation

According to the principles of polymer physics, the degree of orientation is defined as the square of the ratio of the long and short axes of the ellipse obtained by laser spot fitting [28]. The degree of orientation of fibrous food was calculated using MATLAB software according to Formula (1): (1)$$\mathrm{Degree}\;\mathrm{of}\;\mathrm{Orientation}={\left(\frac{L_{L}}{L_{S}}\right)}^{2}$$
where *L_L_* and *L_S_* are the lengths of the long and short axes of the ellipse, respectively.

### 2.7. Verification of the Determination Method

#### 2.7.1. Degree of Orientation Determination

The laser spot images of extruded pork jerky, beef jerky, chicken jerky, and duck jerky were obtained using the constructed laser transmission imaging system. The ellipse fitting effects and the degree of orientation between these samples were compared.

#### 2.7.2. Pseudocolour Maps Acquisition

The original laser spot images of each sample were processed using ImageJ version 1.52v (National Institutes of Health, Bethesda, MD, USA) to obtain the pseudocolour maps. The reliability of the method for determining the degree of orientation was verified by comparing the characteristics of different pseudocolour maps.

#### 2.7.3. Microstructural Characteristics 

SEM (Quanta^TM^ 250 FEG, Field Electron and Ion Co., Hillsboro, OR, USA) was used to observe the microstructure of the jerky samples. The samples were placed on a special holder, fixed, and then coated with AuPd. The micrographs were then observed at a magnification of ×500. The reliability of the method for determining the degree of orientation was evaluated according to the microstructural characteristics of the samples tested.

#### 2.7.4. Textural Properties 

The textural properties of the extruded jerky were determined using a TMS-Pro texture analyser (Food Technology Inc., Sterling, VA, USA). Each sample was placed on the special test board parallel or perpendicular to a blade probe (432-018) at a speed of 1 mm/s until 75% of the cutting depth was reached. Transverse and longitudinal shear forces were determined from the force-versus-time curves. The degree of organisation, a commonly used index to determine the orientation of a fibre structure, was then obtained from the ratio of transverse shear force to longitudinal shear force. 

### 2.8. Statistical Analysis

All determinations were repeated at least 10 times, and the final results are expressed as the mean ± standard deviation. Significant differences (*p* < 0.01) in the results of the textural properties and degree of orientation were analysed using SPSS version 27.0 (IBM Co., New York, NY, USA). The coefficients of the variation data were obtained to investigate the reliability of the orientation results. To further compare the results of the degree of organisation and orientation, the corresponding data were normalised using the mean normalisation method, and their correlations were analysed using the Spearman correlation analysis method.

## 3. Results

### 3.1. Acquired Laser Spot Images

The original laser spot image of the extruded pork jerky is shown in Figure 7. Close examination of the laser spot image reveals a granular, irregularly distributed red scattering spot, known as laser speckle noise. The presence of this noise inevitably reduces the clarity of the image and seriously affects the reliability of the internal information extracted from the laser spot image. Therefore, further image processing is required to obtain reliable orientation results [41].

### 3.2. Results of Laser Spot Image Processing 

#### 3.2.1. Gaussian Filter Denoising

The blurring and smoothing effects of the laser spot image by the Gaussian filter template are directly related to the reliability of the subsequent ellipse fitting results. Figure 8 shows a comparison of the image results of pork jerky samples using different sizes of Gaussian filter templates (5 × 5, 7 × 7, and 9 × 9). The edges of the laser spot are blurred and smoothed as the size of the Gaussian filter template increases. If the template used is extremely large or small, the purpose of the ellipse fitting process will not be effective or completed successfully. As the differences in the blurring and smoothing effects among Figure 8a–c are not clearly visible, further image processing steps are required to progressively determine the optimum Gaussian filter template for laser spot image processing.

#### 3.2.2. Edge Enhancement

The original laser spot image successfully processed using the Gaussian and Edge filter templates is shown in Figure 9. The different sizes of Gaussian filter templates combined with the 3 × 3 Sobel edge filter template were used to process laser spot images of pork jerky samples (Figure 9a–c). The results show that all images are characterised by no visible background noise and relatively clear elliptical spot boundaries. The laser spot images shown in Figure 9d–f were also processed with different sizes of Gaussian filter templates but combined with the 5 × 5 Sobel edge filter template, resulting in a lot of background scatter noise and a wide elliptical spot boundary. This made it very difficult to select the exact locations of the pixels during the subsequent ellipse fitting process. Therefore, the 3 × 3 Sobel edge filter template is confirmed to be the better choice when dealing with laser spot images.

Based on the confirmation of the use of 3 × 3 Sobel edge filter templates, the appropriate size of Gaussian filter templates can be selected. Compared to the laser spot image processed with the 9 × 9 Gaussian filter template (Figure 9c), the boundary between the red and black regions of the red ellipse edges obtained with the 3 × 3 Gaussian filter template was bright and clear, as shown in Figure 9a. Since there were no significant differences between Figure 9a,b, both the 5 × 5 and 7 × 7 Gaussian filter templates were combined with the 3 × 3 Sobel edge filter template in subsequent image processing steps until the optimal Gaussian filter template was confirmed.

#### 3.2.3. Morphological Smoothing

The images obtained above were then processed by morphological smoothing to produce the laser spot images shown in Figure 10. Compared to Figure 9, the tiny gaps in the images shown in Figure 10 are completely filled while preserving the original image feature information, and the discontinuous pixels are joined together, dramatically increasing the sharpness of the elliptical laser spots.

#### 3.2.4. Binarisation Treatment

Based on the results shown in Figure 10, the images were further processed using a binarisation treatment method. As shown in Figure 11, the binarised images contained only black and white background colours. The colour contrast at the elliptical patch boundary was significant, making it easy to fit the ellipses to calculate the degree of orientation.

#### 3.2.5. Ellipse Fitting

After the image processing described in Section 3.2.1, Section 3.2.2, Section 3.2.3 and Section 3.2.4, the boundaries of the laser spots were relatively clear, and an ellipse could be fitted directly from the outermost coordinate points of the image. The results of the fitted ellipses are shown as the solid red line in Figure 12. 

### 3.3. Results of the Degree of Orientation 

For foods with a non-oriented fibre structure, the texture is uniform in all directions, showing a general isotropy, meaning that the results of their degree of orientation tend to be 1. On the other hand, for foods with a well-organised fibre structure, the lengths of the long and short axes of the fitted ellipses are quite different. The considerable lengths of the long axes and the minimal lengths of the short axes indicate favourable orientation results for extruded jerky. Based on the fitted ellipses, it is easy to calculate the degree of orientation of the extruded pork jerky, which is in the range of 1.30–1.51 according to Equation (1).

In short, a new method for determining the degree of orientation has been successfully developed. To verify the reliability of this method, more jerky samples and conventional research methods were used in the validation experiments.

### 3.4. Validation of the Orientation Determination Method 

#### 3.4.1. Orientation Results for Extruded Jerky 

Figure 13 shows the results of the fitted ellipses for extruded jerky samples. According to the determination method developed in our research, the calculated results of the degree of orientation for extruded beef jerky, pork jerky, chicken jerky, and duck jerky are 1.56, 1.40, 1.25, and 1.20, respectively. The results of the calculated data approach 1 as the fitted curves of the laser spots gradually move from an ellipse to a circle. The comparative analysis of Figure 13 shows that the orientation effects are in the ranking of beef jerky > pork jerky > chicken jerky > duck jerky, which exactly matches the results of the degree of orientation data. This result is also consistent with empirical perceptions of the effects of fibre arrangement in these extruded jerky products.

#### 3.4.2. Comparison of Pseudocolour Map Differences 

The pseudocolour maps obtained directly from the raw laser spot images using commercial software ImageJ version 1.52v are shown in Figure 14, and significant differences among the extruded jerky types can be easily observed. The peripheral contour of the laser spot formed by the red pseudocolour portion shows an elliptical shape, confirming the feasibility of the orientation determination method from the underlying principle. In addition, the order of proximity to the ellipse of the pseudocolour maps is consistent with the degree of orientation results, confirming that this determination method is basically reliable.

#### 3.4.3. Comparison of Structural Characteristics 

The structural characteristics of the extruded jerky are shown in Figure 15 to further validate the reasonableness of the above ranking of the degree of orientation of the extruded jerky samples. Without any data support, it still shows the basic trend of fibre orientation. Overall, the fibre microstructure in the extruded beef jerky appears to have the best orientation effects (Figure 15a), although ingredients such as soya bean isolate protein and potato starch, adhering to the surface of the samples, may affect the reliability of the degree of orientation results. The micrographs of the extruded pork and chicken jerky in Figure 15b,c show the next best fibre orientation effects, whereas the duck jerky in Figure 15d appears to have poor fibre orientation. There is clearly a significant difference in the orientation effects between Figure 15a,d. These results confirm that the ranking trend in the degree of orientation (beef jerky > pork jerky > chicken jerky > duck jerky) is reasonable from another point of view, indicating that the method for determining the degree of orientation is reliable.

#### 3.4.4. Comparison and Analysis of Textural Properties

The results of the textural properties of the extruded jerky are shown in Table 1. The average values of transverse and longitudinal shear strength are highest for beef and duck jerky, respectively, while chicken jerky has the lowest transverse and longitudinal shear strength. The strength and degree of cross-linking of the different types of meat fibres are responsible for these results. There is no significant difference in the degree of organisation between beef jerky and pork jerky, nor between pork jerky and chicken jerky, except for the corresponding results for duck jerky. However, the results of the significance analysis marked with letters a, b, and c indicate that there is a highly significant difference in orientation among beef jerky, pork jerky, and chicken jerky samples (*p* < 0.01). This suggests that the method for determining the degree of orientation can effectively discriminate these types of extruded jerky, although no significant difference in orientation can be seen between chicken jerky and duck jerky. In addition, the coefficients of the degree of organisation for extruded jerky range from 8.08 to 10.39, which is significantly higher than the corresponding range of the degree of orientation results (4.20–4.82), indicating the reliability of this orientation determination method.

The ascending order of the mean for the degree of organisation is beef jerky > pork jerky > chicken jerky > duck jerky, which corresponds to the ranking of the data for the degree of orientation (Figure 16a). As shown in Figure 16b, the results after the normalisation treatment show significantly similar trends, and the results of the data correlation analysis show that the degree of organisation and the degree of orientation have a significantly strong positive correlation with each other (r = 0.68, in the range of 0.6–0.8, *p* < 0.01), confirming that this newly developed method for determining the degree of orientation is reliable.

## 4. Discussion

### 4.1. Comparing Findings with Similar Studies

This paper presents a reliable approach to quantifying the degree of orientation in fibrous foods using laser transmission and computer vision. The current methods for determining the degree of orientation in fibrous foods are limited. Most studies use a similar definition, referred to as degree of organisation, to describe fibre structure [8,9,10]. The degree of organisation results are derived from the textural properties, as mentioned in Section 2.7.4. However, this mechanical method is destructive and tends to lose important internal structural information due to the relatively rough data acquisition process, resulting in insufficiently reliable results.

Ranasinghesagara et al. [28] introduced a non-destructive photon migration method to determine the degree of fibre orientation in meat analogues. However, our image acquisition systems use different instrumentation details, and our orientation calculation algorithm is completely independent. We have further validated the orientation determination method from a number of perspectives, with a greater focus on the correlations between the organisation results and the orientation data.

In addition, computer image recognition methods, including neural image analysis techniques using deep learning convolutional neural networks, can only provide surface information rather than the internal structural information required to calculate the degree of orientation.

### 4.2. Comparing Orientation Determination Methods

In the field of polymer physics, several methods have been established to determine orientation, such as in-plane birefringence, sound wave propagation, Raman spectroscopy, infrared spectroscopy, wide-angle X-ray diffraction, polarised fluorescence spectroscopy, nuclear magnetic resonance spectroscopy, small-angle X-ray scattering, small-angle neutron scattering, and small-angle light scattering [42]. Since polymers consist of different orientated units, such as chain links, segments, whole polymer chains, or crystalline regions, the results obtained by different methods may only represent the orientation of specific units, and their significance varies accordingly.

Because many oriented foods do not have stable refractive indices, the birefringence method often fails to produce a clear Becke line. We encountered this situation when attempting to measure the orientation of extruded and natural jerky using the in-plane birefringence method. The sound wave propagation method is primarily used to measure long polymer fibres and is almost impossible to apply to food samples. In addition, spectroscopic methods require different instruments, which can be very costly, and some methods require meticulous sample preparation, which significantly limits their applicability.

We are currently comparing this method with other techniques, including infrared dichroism and small-angle X-ray scattering, which can simultaneously measure the orientation of crystalline and non-crystalline regions. Our aim is to establish a mathematical relationship between the results of these widely used methods in polymer physics and the newly developed method. This will certainly increase the importance of our method in the field of polymer physics.

### 4.3. Advantages and Limitations of Our Method

The validation experiments for this orientation determination method are carried out using extruded jerky as an illustrative example. However, it is important to note that this method has a wide range of applications in other fibrous foods, including meat analogues, dried mushrooms, extruded high-protein pasta, and more. These results will be detailed in a forthcoming study.

This method has several key attributes, including speed, accuracy, simplicity, and non-destructiveness, in line with the principles of laser transmission and computer vision. It is worth noting that our method has minimal sample size requirements, with the laser spot area not exceeding 0.2 cm², allowing for the use of very small fibre samples to determine the degree of orientation. Furthermore, our method does not require any special sample preparation and can facilitate on-line orientation measurements through computer vision analysis.

Despite the advantages of this newly developed method, there are certain limitations. As shown in Formula (1), the degree of orientation is calculated based on the lengths of the long and short axes of the ellipse, which differs from the determination principles of other methods in the field of polymer physics. Consequently, our results cannot be directly compared with the degree of orientation in high-polymer materials. In addition, this method cannot be applied to completely opaque, transparent, or highly reflective fibrous foods. Fortunately, the majority of fibrous foods studied do not have these characteristics and are therefore suitable for use with this method.

## 5. Conclusions

The current study successfully developed a new method for determining the degree of orientation of fibrous foods by combining laser transmission and computer vision. The laser transmission imaging system was constructed using a red laser beam to irradiate the fibre axis direction of fibrous samples. Original laser spot images were captured using a digital camera, and the obtained laser spot images were then subjected to further image processing operations, including: (1) denoising using a 7 × 7 Gaussian filter template; (2) enhancing image edges using a 3 × 3 Sobel edge filter template; (3) connecting image break points with morphological smoothing operation; (4) highlighting image edge coordinates with a binarisation treatment; and (5) fitting elliptical laser spots using the least squares method. Finally, the fitted ellipse obtained from the coordinates shifted inward by 20 pixels was selected as the optimal basis for calculating the degree of orientation of fibrous foods.

The results of the degree of orientation for extruded beef jerky, pork jerky, chicken jerky, and duck jerky are 1.56, 1.40, 1.25, and 1.20, respectively, showing a reasonable ranking of orientation effects as follows: beef jerky > pork jerky > chicken jerky > duck jerky. The reasonableness of this ranking is verified by the graphs shown in the pseudocolour maps and also by the microstructural characteristics. The significance of the differences shows that this method can be used to identify different types of fibrous foods. In addition, the relatively small coefficients of variation and the strong positive correlation between the results of the degree of organisation and orientation confirm the reliability of this newly developed method.

This method has a wide range of potential applications in various fibrous foods and offers numerous advantages, including reliability, speed, accuracy, simplicity, non-destructiveness, and minimal sample size requirements, etc. It helps to fill the gap in related research areas, as there are few similar studies focusing on orientation determination in the food research field. The aforementioned characteristics of this newly developed method hold great promise for future implementation in the field of online detection.

## Figures and Tables

**Figure 1 foods-12-03541-f001:**
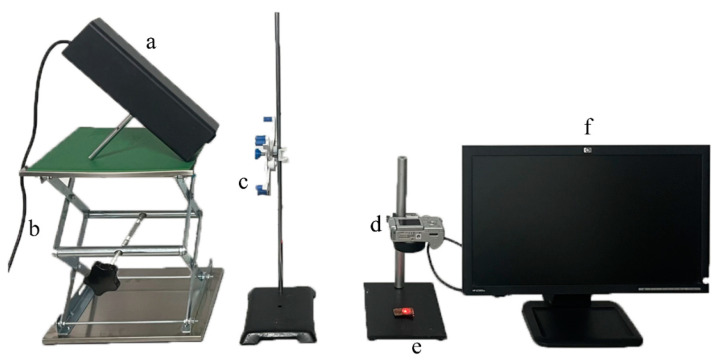
Main components of the laser transmission imaging system: laser transmitter (**a**), lifting platform (**b**), neutral filter (**c**), digital camera (**d**), sample holder (**e**), and computer (**f**).

**Figure 2 foods-12-03541-f002:**
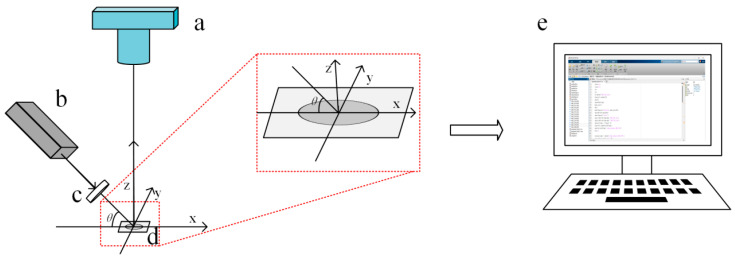
Laser spot image acquisition process based on digital camera (**a**), laser transmitter (**b**), neutral filter (**c**), fibre sample (**d**), and computer (**e**).

**Figure 3 foods-12-03541-f003:**
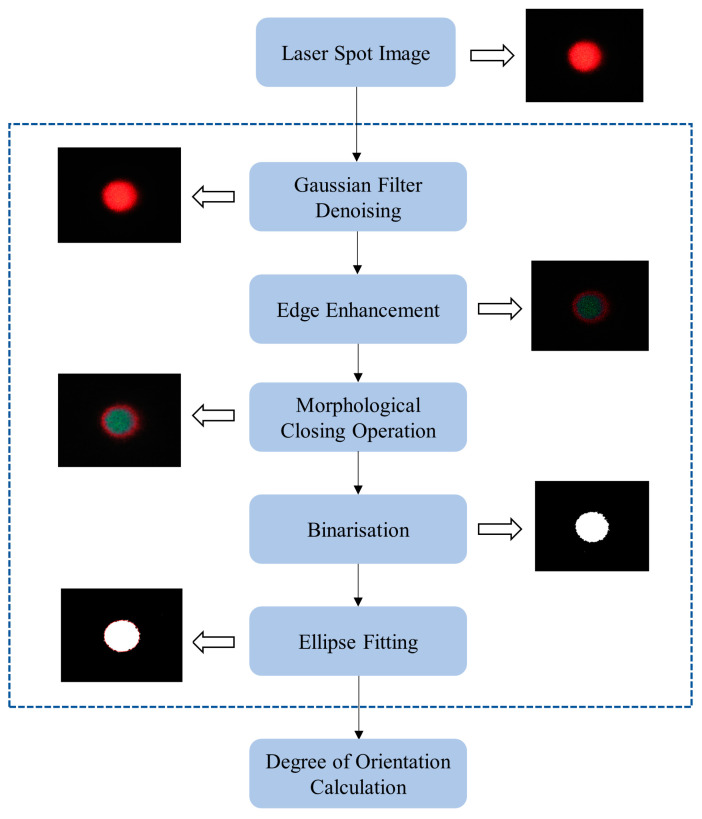
Scheme of the image processing steps.

**Figure 4 foods-12-03541-f004:**
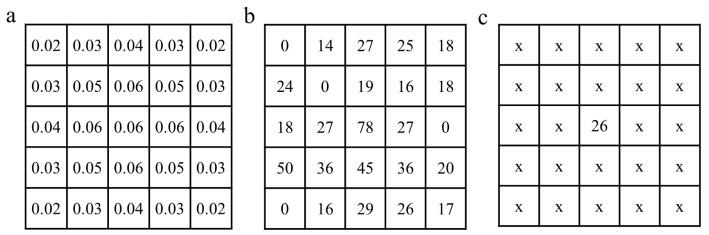
Gaussian filter template with a size of 5 × 5 (**a**), and pixel values of the image before (**b**) and after (**c**) the Gaussian filter denoising process. The “x” in figure (**c**) does not refer specifically to any pixel value.

**Figure 5 foods-12-03541-f005:**
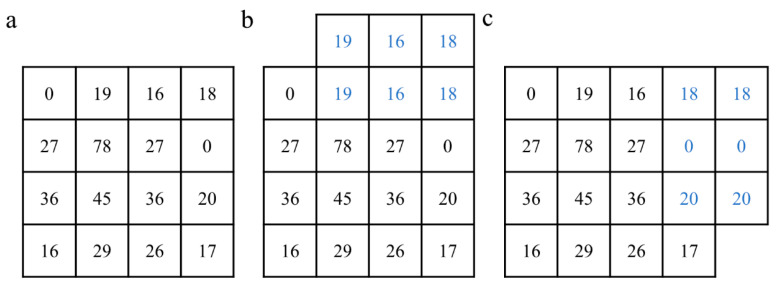
Pixel values of the original image (**a**) and theoretical results after top (**b**) and right (**c**) edge enhancement.

**Figure 6 foods-12-03541-f006:**
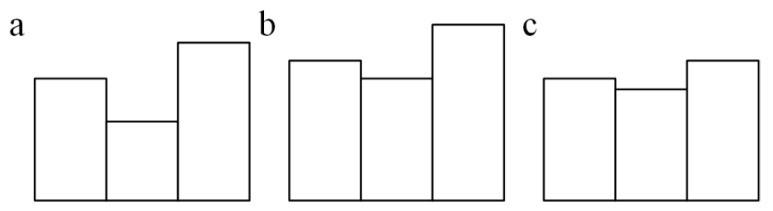
Initial image (**a**), expanded image (**b**), and the image processed by the expansion operation followed by the erosion operation (**c**).

**Figure 7 foods-12-03541-f007:**
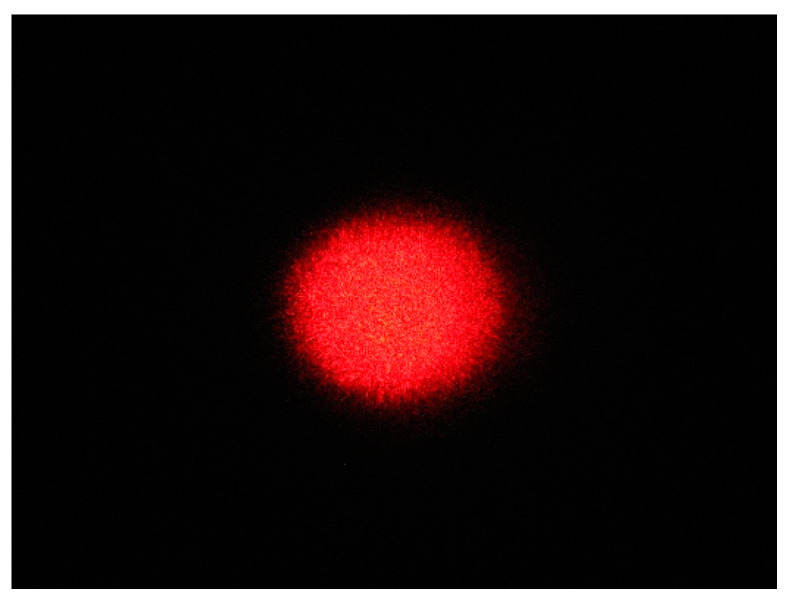
Image of the original laser spot for extruded pork jerky.

**Figure 8 foods-12-03541-f008:**
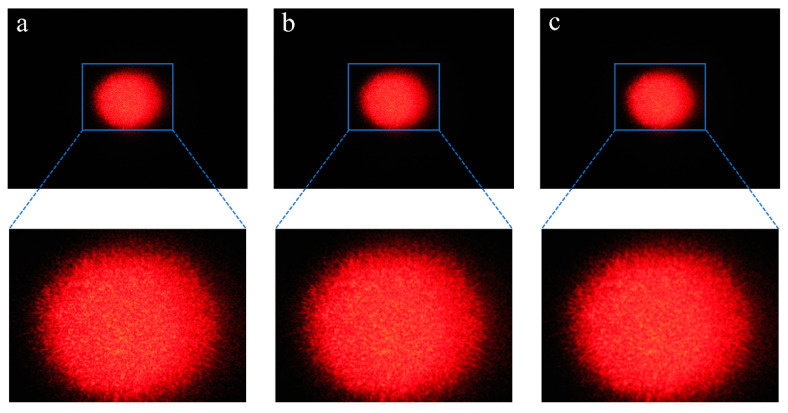
Images processed with Gaussian filter templates with sizes of 5 × 5 (**a**), 7 × 7 (**b**), and 9 × 9 (**c**). The images connected by dotted lines are partially enlarged images.

**Figure 9 foods-12-03541-f009:**
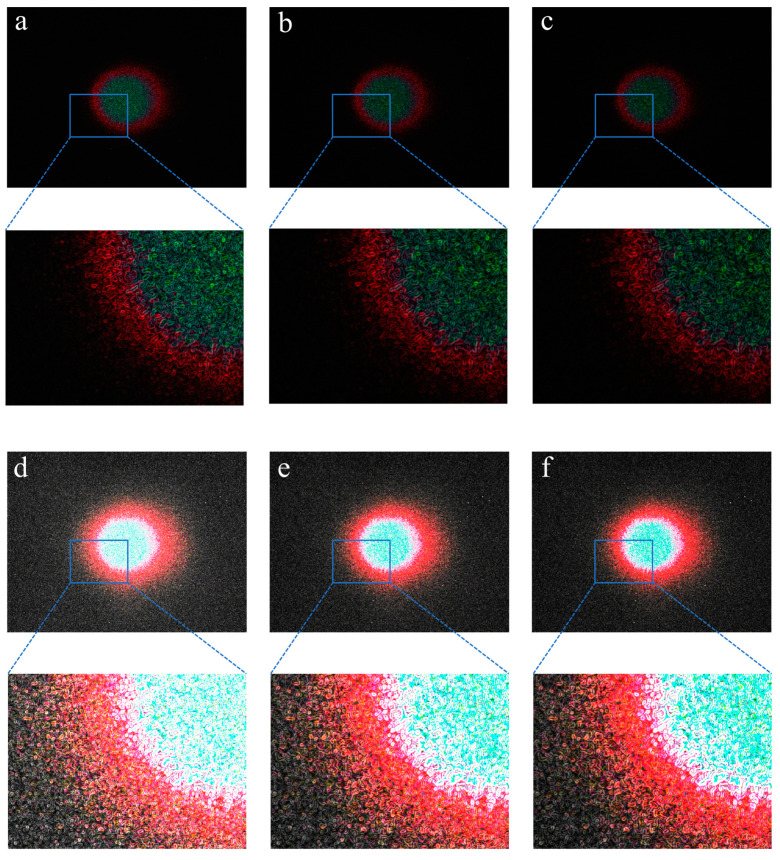
Laser spot images processed with Gaussian and edge filter templates. (**a**–**c**) Images processed with 5 × 5, 7 × 7, and 9 × 9 Gaussian filter templates combined with the 3 × 3 Sobel edge filter template, respectively. (**d**–**f**) Images processed with 5 × 5, 7 × 7, and 9 × 9 Gaussian filter templates combined with the 5 × 5 Sobel edge filter template, respectively. The images connected by dotted lines are partially enlarged images.

**Figure 10 foods-12-03541-f010:**
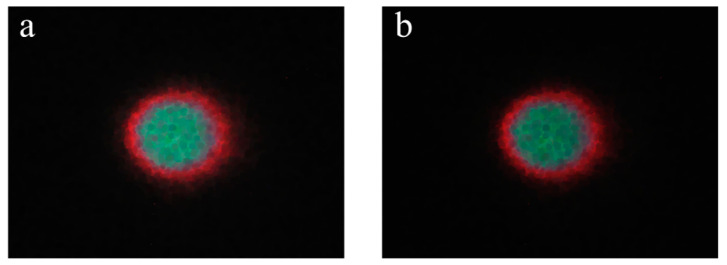
Laser spot images obtained with 5 × 5 (**a**) and 7 × 7 (**b**) Gaussian filter templates, followed by the 3 × 3 Sobel edge filter template and the morphological smoothing operation.

**Figure 11 foods-12-03541-f011:**
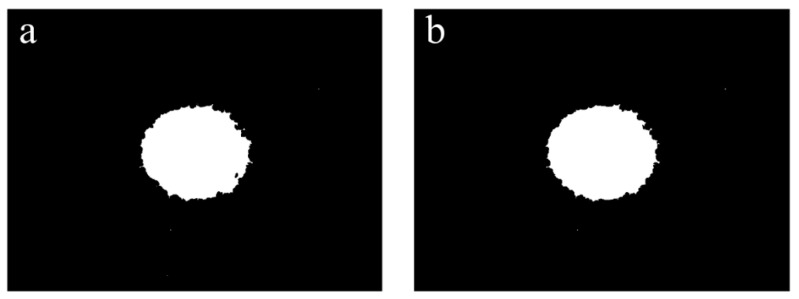
Laser spot images processed with 5 × 5 (**a**) and 7 × 7 (**b**) Gaussian filter templates, respectively, followed by the operations of the 3 × 3 Sobel edge filter template, the morphological smoothing operation, and the binarisation treatment.

**Figure 12 foods-12-03541-f012:**
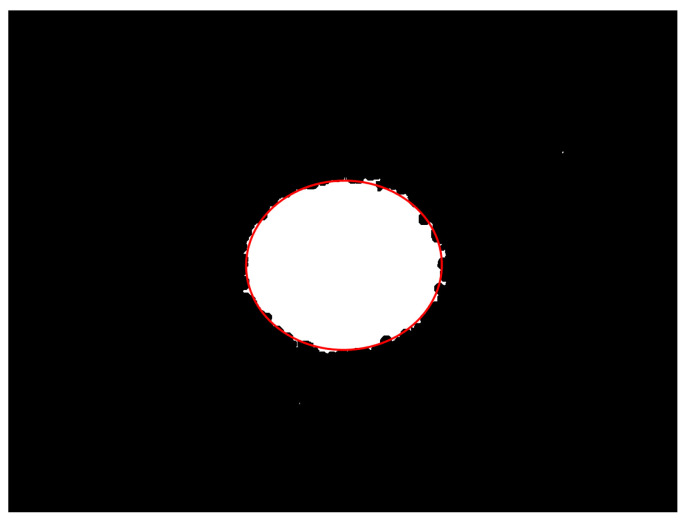
Fitted ellipse obtained by fitting to the outermost coordinates of the laser spot.

**Figure 13 foods-12-03541-f013:**
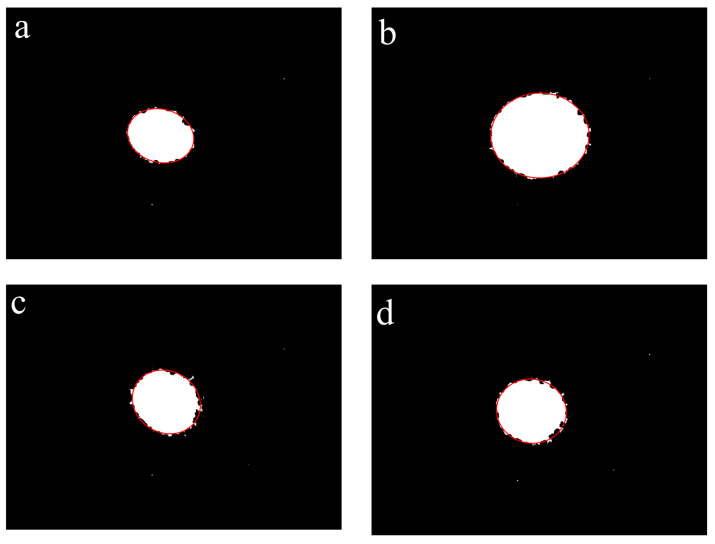
Fitted ellipses of the laser spot images for extruded beef jerky (**a**), pork jerky (**b**), chicken jerky (**c**), and duck jerky (**d**).

**Figure 14 foods-12-03541-f014:**
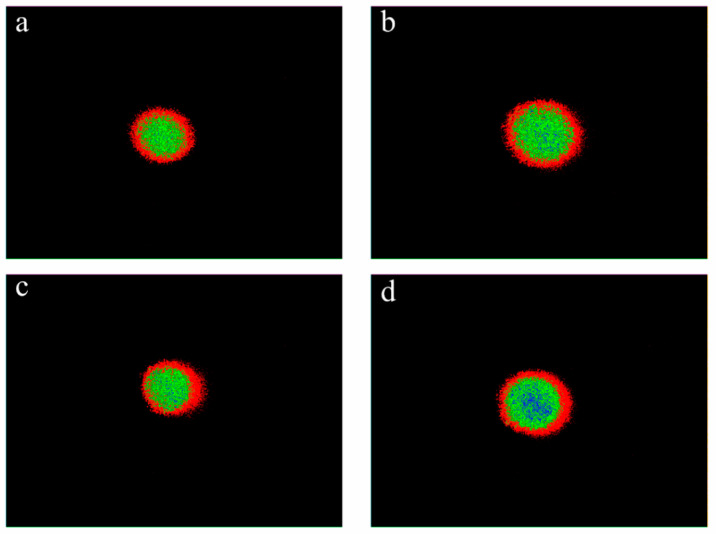
Pseudocolour maps extracted from laser spot images for extruded beef jerky (**a**), pork jerky (**b**), chicken jerky (**c**), and duck jerky (**d**).

**Figure 15 foods-12-03541-f015:**
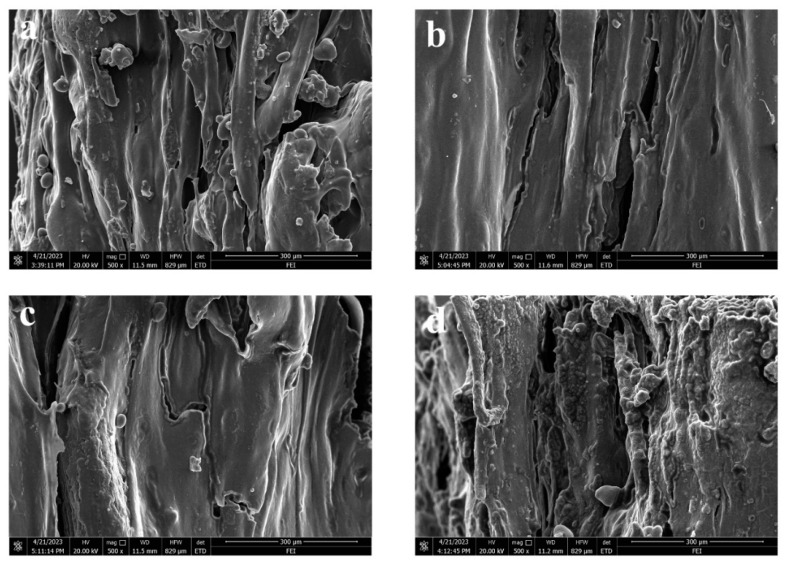
Structural micrographs for extruded beef jerky (**a**), pork jerky (**b**), chicken jerky (**c**), and duck jerky (**d**).

**Figure 16 foods-12-03541-f016:**
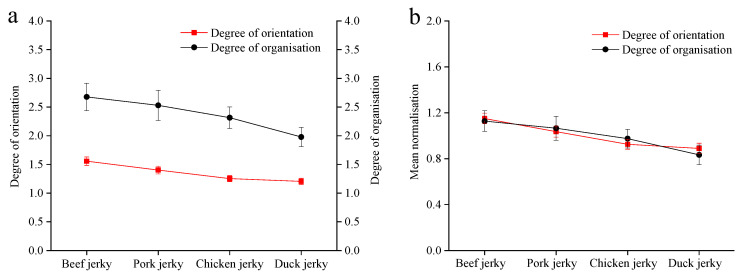
Trend of variation in the initial data for degree of organisation and orientation (**a**) and the corresponding data after the normalisation treatment (**b**).

**Table 1 foods-12-03541-t001:** Comparison of textural properties of extruded jerky.

Extruded Samples	Descriptive Indicators	Transverse Shear Force (N)	Longitudinal Shear Force (N)	Degree of Organisation	Degree of Orientation
Beef jerky	Variable amplitude	91.20–113.17	30.30–43.91	2.43–3.32	1.42–1.64
	Average value	100.53 ± 6.62 a	37.81 ± 3.69 bc	2.68 ± 0.24 a	1.56 ± 0.07 a
	Coefficient of variation (%)	6.59	9.76	8.99	4.66
Pork jerky	Variable amplitude	93.91–119.19	32.70–51.62	2.23–3.08	1.30–1.51
	Average value	102.60 ± 7.49 a	41.02 ± 5.40 b	2.53 ± 0.26 ab	1.40 ± 0.07 b
	Coefficient of variation (%)	7.30	13.17	10.39	4.82
Chicken jerky	Variable amplitude	61.55–88.44	23.78–41.60	2.06–2.63	1.16–1.32
	Average value	78.42 ± 8.10 b	34.22 ± 5.16 c	2.31 ± 0.19 b	1.25 ± 0.05 c
	Coefficient of variation (%)	10.34	15.07	8.08	4.20
Duck jerky	Variable amplitude	83.22–110.16	40.42–58.40	1.74–2.37	1.12–1.29
	Average value	100.28 ± 7.64 a	50.99 ± 5.26 a	1.98 ± 0.17 c	1.20 ± 0.05 c
	Coefficient of variation (%)	7.62	10.32	8.5	4.51

## Data Availability

The data presented in this study are available upon request from the corresponding author.

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
