# Peer review of "A Reliable Method for Determining the Degree of Orientation of Fibrous Foods Using Laser Transmission and Computer Vision"

_foods, 2023, doi:10.3390/foods12193541_

Round 1
Reviewer 1 Report
The Abstract should also state the problem being solved by this study.
Discuss the significance of 20 pixel-shifting. What is the case if there is only 19 pixels shifted or how about 21 pixels? Provide technical supports.
List the potential contributions of this study at the end part of Introduction section.
Why Gaussian filter denoising was used and not other filters?
The number of samples per material used should be indicated.
ANOVA with post hoc tests is highly recommended as there are three replicates conducted in this study and provide error bar plots with groupings to properly elucidate the consistency of findings.
The Discussion section was very poorly written. There is a need to provide comparisons of findings with other studies which used different vision system and the corresponding metrics. It would also be better if there is discussion about the novelty of this approach and its limitations.
The changes in microstructure which represent the reliability of the technique was not properly elucidated both in the Results and Discussion sections.
Requires minor editing of English language.
Reviewer 2 Report
The title of this study is: A Reliable Method for Determining the Degree of Orientation of Fibrous Foods Using Laser Transmission and Computer Vision.
I commented on the manuscript and the comments are presented below:
Part: Introduction.
The Introduction to the study is broad and does end with a clearly stated purpose or goals that the Authors wish to pursue.
Part: Material and methods
The Methods section provides the reader with enough information to repeat the experiments conducted. Only basic statistical analysis was performed. Advanced statistical analysis of the obtained results was not performed. An advanced statistical analysis should be carried out for the results obtained as a result of the measurements carried out. You can determine the strength of the influence of a particular parameter on the variance of the system. At the same time, correlation relationships between the determined parameters can be determined.
Part: Results and discussion
For the most part the Results section is well structured.
In the Discussion chapter, there is no full comparison and confrontation with the research of other authors in this area. The results were not fully discussed. A full discussion of the results obtained with other work in this field should be carried out in more aspects. I suggest supplementing the Chapter with additional information.
Part: Conclusion
The Conclusions chapter contains information obtained after conducting experiments but performing only base statistical analyzes and were no comparison and confrontation with the research of other authors in this area.
Part: References.
The literature used is appropriate but should be supplementing about the items from the last years of publication about similar problem.
Reviewer 3 Report
The work developed an original method using laser transmission and digital image analysis to quantitatively characterize the structure of food fibers. The degree of ordering the fiber structure significantly affects the textural properties and the level of consumer acceptance of fibrous food, such as extruded dried beef, dried pork, dried chicken and dried duck. The authors state that small coefficients of variation and a strong correlation between the degree of organization and the degree of orientation confirm the effectiveness and credibility of the proposed method. The authors also emphasized the utilitarian nature of the proposed identification technique and its universality.
The proposed method is interesting and fits into the scientific space represented by your journal.
However, it requires a few clarifications, additions and corrections of a syntactic nature:
- The texture of the meat fibers is an important quality characteristic of the product. However, there are other important parameters, such as marbling, physical parameters, semantic features, etc. Is the identification of the character of the texture a sufficient criterion for the quality assessment of meat?
- How does the proposed identification method correspond with other computer methods used to support digital image analysis, eg neural image analysis techniques using deep-learned convolutional neural networks?
- Paragraph 240: LL i Ls na LL oraz Ls
Round 2
Reviewer 2 Report
The authors referred to the comments from the previous review for the manuscript titled: A Reliable Method for Determining the Degree of Orientation of Fibrous Foods Using Laser Transmission and Computer Vision. I accept explanations. In the future, I suggest using more precise describing relationships between the parameters studied. They supplemented the discussion with a new literature data strengthens the message and importance of information in the manuscript.